# Exposure of Humans to Attacks by Deer Keds and Consequences of Their Bites—A Case Report with Environmental Background

**DOI:** 10.3390/insects11120859

**Published:** 2020-12-03

**Authors:** Weronika Maślanko, Katarzyna Bartosik, Magdalena Raszewska-Famielec, Ewelina Szwaj, Marek Asman

**Affiliations:** 1Department of Animal Ethology and Wildlife Management, Faculty of Animal Sciences and Bioeconomy, University of Life Sciences in Lublin, Akademicka 13 St., 20-950 Lublin, Poland; 2Chair and Department of Biology and Parasitology, Faculty of Health Sciences, Medical University of Lublin, Radziwiłłowska 11 St., 20-080 Lublin, Poland; katarzyna.bartosik@umlub.pl; 3Department of Cosmetology, Faculty of Physical Education and Health, University of Physical Education, Akademicka 2 St., 21-500 Biała Podlaska, Poland; magdalena.raszewska-famielec@awf-bp.edu.pl; 4NZOZ Med-Laser Dermatology Clinic, Młyńska 14A St., 20-406 Lublin, Poland; 5Ignacy Jan Paderewski Primary School Number 43 in Lublin, Śliwińskiego 5 St., 20-861 Lublin, Poland; szwajewelina@gmail.com; 6Department of Parasitology, Faculty of Pharmaceutical Sciences in Sosnowiec, Medical University of Silesia, Jedności 8 St., 41-200 Sosnowiec, Poland; masman@sum.edu.pl

**Keywords:** deer ked bites, deer ked dermatitis, *Lipoptena cervi*, wild cervids, ectoparasites, blood-sucking arthropods

## Abstract

**Simple Summary:**

*Lipoptena* species, also named the deer ked or deer fly, are commonly encountered in temperate areas of Europe, northern China, and North America. Although wild animals seem to be the preferred hosts of these parasitic arthropods, it is increasingly being noted that humans are also directly threatened by their bites. Skin lesions evolving after *Lipoptena* bites are painful and often lead to the development of inflammation of the skin. *Lipoptena* species also pose a threat to the health of the host by transferring pathogenic factors, e.g., *Bartonella schoenbuchensis*, *Borrelia burgdorferi*, and *Anaplasma phagocytophilum*. For this reason, knowledge of symptoms caused by *Lipoptena* bites is worth promoting among not only scientists but also the general public.

**Abstract:**

Insects of the genus *Lipoptena,* e.g., *Lipoptena cervi* and *Lipoptena fortisetosa*, are hematophagic ectoparasites mainly attacking deer, roe deer, moose, horses, and cattle. Humans may also be incidental hosts for these insects. The species are vectors of numerous pathogens, including *Bartonella schoenbuchensis*, *Borrelia burgdorferi*, and *Anaplasma phagocytophilum*. Due to the short time of feeding on humans, usually painless bites, and an initially small trace at the site of the bite, the symptoms reported by the patient may not be associated with deer ked infestation and infection with pathogens transmitted by these arthropods. The aim of the study was to describe the consequences of *L. cervi* bites in humans with detailed documentation of the development of skin lesions. The knowledge about skin lesions arising after deer ked bites may be useful in clinical practice for monitoring patients for the presence of pathogens transmitted by the parasites.

## 1. Introduction

Arthropods from the Hippoboscidae family are obligate parasites of mammals and birds. The Hippoboscidae family includes three subfamilies: Ornithomyinae, Hippoboscinae, and Lipopteninae. Although wild animals seem to be the preferred hosts of these parasitic arthropods, it is increasingly being noted that humans are also directly threatened by their bites. *Lipoptena* species (e.g., *Lipoptena cervi* Linnaeus, 1758 and *Lipoptena fortisetosa* Maa, 1965), named the deer keds or deer flies, are commonly encountered in temperate areas of Europe, Siberia, northern China, and North America [1,2,3]. Both male and female adults are the only developmental stages that feed on blood. They parasitize mainly on wild ruminants, e.g., moose and deer burrowing through the fur and sucking the blood of the animal host.

Despite the relatively high veterinary and medical importance and increasing abundance of *Lipoptena* in Europe, the biology of these species requires further research [3,4]. *Lipoptena cervi* is a common species in most European countries; however, it was officially recorded on cervids in Turkey in 2007–2010 [5] and on European roe deer (*Capreolus capreolus*) in the north-eastern region of Romania in 2014 [6]. In Poland, the species is distributed throughout the country [7], and its dynamic spread and the nuisance caused to people seem to be increasing in recent years. A massive attack of *L. cervi* in Poland was documented as early as 1975 in the Niepołomicka Forest, where attacks by up to 20–30 adult individuals per human were observed within 5 min [8]. Humans are accidental hosts for *Lipoptena* species, as they cannot reproduce after feeding on humans. For this reason, this is even considered an ecological trap [9,10], because when the parasite settles on the host, it loses its wings.

*Lipoptena fortisetosa* is a species typical of Asian deer: the Siberian roe deer (*Capreolus pygargus*) and the eastern deer (*Cervus nippon*) [11,12]. The first reports of its presence in Europe came in the 1960s [7], and its progressive range expansion has been observed over the past years [13,14]. In Poland, the first observations of *L. fortisetosa* were made in Lower Silesia at the end of the 1980s [15]. Earlier, in extensive studies carried out by Kadulski from 1973–1980, only *L. cervi* was recorded in Poland (cited after [7]). *L. fortisetosa* was observed again in 2007–2012 on red deer and roe deer in the north of the country [16,17] and in natural habitats located in north-eastern and southern Poland, including the Polish part of the Tatra Mountains [18,19].

Depending on the geographical region, climate, and ecological conditions, the prevalence and severity of invasion of ectoparasites vary significantly, e.g., in the case of *L. cervi*, it mainly depends on the host species [20] and seasonal differences. The highest prevalence is most often noted in autumn and winter [21,22]. Depending on the geographical region, there are differences among the main hosts of *Lipoptena* species. For example, moose is the main host for *L. cervi* in Scandinavia, and *Cervus elaphus* and *Capreolus capreolus* are indicated as important hosts of this species in different regions of Europe [2,14,16,23,24,25] (own observations). In contrast, there are no clear data on the dominant host of *L. fortisetosa*, although it is suggested that this role is probably played by roe deer [7].

A single host may be infested by up to 17,000 *Lipoptena* specimens, and their most frequent and most intense infestation was found in moose, with the prevalence of infection reaching even 80–100% [4,26,27]. Such massive invasions of *L. cervi* in Scandinavia were regarded as one of the factors causing a deadly disease called the “hair loss syndrome” in moose [28]. The increase in the number of *L. cervi* may correlate with the increase in the population size of moose, which plays the role of its main host in Finland [4,23]. Given the ban imposed on moose hunting in Poland in 2001 resulting in a gradual increase in the number of moose, deer ked attacks may also become a problem in the Polish moose population soon.

*Lipoptena* species pose a threat to the health of the host as potential vectors of numerous pathogens. The greatest epidemiological importance is attributed to *Bartonella schoenbuchensis, Borrelia burgdorferi*, and *Anaplasma phagocytophilum* [29,30,31,32,33,34] as well as *Trypanosoma* (*Megatrypanum*) spp. [14,35]. People entering natural habitats of *Lipoptena* spp. are exposed to their attacks and the subsequent health consequences; therefore, knowledge of the clinical picture of *Lipoptena* bites may be helpful in proper diagnosis and further therapeutic management. In the literature on the subject, publications describing skin lesions in patients rarely contain photographic documentation, valuable in the diagnostic process. In our opinion, knowledge of symptoms caused by *Lipoptena* bites is worth promoting among not only scientists but also the general public.

## 2. Materials and Methods

Skin lesions resulting from *Lipoptena* bites on a member of our research team were documented using a camera in a HUAWEI Y6 Prime 2018 smartphone (Huawei Technologies Co., Ltd., Shenzhen, China) and described by a specialist in dermatology. Identification of the insect species and the developmental stage was carried out using an OLYMPUS SZX16 (Olympus, Tokyo, Japan) stereoscopic microscope and the key to the identification of arthropod species by Borowiec [1].

## 3. Results

### 3.1. Environmental Background and Circumstances of Lipoptena cervi Bites

A 35-year-old woman was walking near a coniferous forest inhabited by numerous wild animals, e.g., deer and roe deer, and moose. Specimens of *Lipoptena* spp. had been observed in this area since spring (grasslands of 2.5 ha, being an ecotone inside coniferous forest complexes in the Puławy State Forest, Eastern Poland). The woman was bitten six times on her left thigh during 45 min contact with the ectoparasite in a car on July afternoon. The parasites got into the car and attacked the woman. The bites by the insect moving on the woman’s legs were painless. It is interesting how the specimen described did not lose its wings after reaching the human host and continued biting. The insect was caught and placed in a plastic container for further investigations.

### 3.2. Case Presentation

The first itching and burning symptoms appeared at night. At 8 a.m., six disseminated urticarial pink papules (the biggest at the time was 15 mm × 25 mm) on an erythematous base were noticed on the thigh (Figure 1A). The deer ked feeding site was visible in the central part of the lesions. Precisely 25 h after biting, the papules became more regular, but with the presence of a severe allergic reaction (Figure 1B).

The skin lesions caused an increasing burning pain. During the following days, the inflammatory reaction was observed to have spread. Three days after the bite, the dermatitis showed a self-limiting tendency (Figure 2). Excoriated erythematous papules appeared in the center of the site of the deer ked bite. The inflammatory reaction resolved over five days, but hyperpigmented papules persisted for three months. In only the biggest site of the deer ked bite, a small field with a clear fluid vesicle was noticed. The *Lipoptena cervi* bites caused the formation of vesicles on the skin, which transformed into small erosions after splitting (Figure 3). The patient did not report any additional systemic symptoms. The patient did not use any ointment to relieve the symptoms, but she only took antihistamines (20 mg of bilastine, and 10 mg of cetirizine once a day) for ten days. The skin lesions were not touched or scratched. The woman underwent diagnostic IFA tests for the detection of *Bartonella* spp. IgM and IgG as well as ELISA tests for the detection of *Borrelia burgdorferi* s.l. IgM and IgG. The results showed abnormalities that required further monitoring.

### 3.3. Insect Identification

During identification and photo works, the wings of the ked dropped off. The insect was identified as a *Lipoptena cervi* female (Figure 4).

## 4. Discussion

Animals and people living or staying for professional and recreational purposes in habitats of parasites are exposed to attacks and infections by ked-transmitted pathogens. The risk is mainly related to the type of environment inhabited by the population of susceptible hosts and is a serious problem for professional groups staying constantly or frequently in the natural environment and threatened directly by zoonotic diseases transmitted by ticks. The problem becomes particularly important in the era of climate change. The increase in average temperatures and the shorter winter periods significantly affect the expansion of ectoparasites, which show high flexibility and the ability to adapt to gradual changes in thermal conditions [22,36,37]. Their spread is also facilitated by the high density of biungulates related to the almost unlimited access to crops and agricultural produce they feed on.

In cervids, anemia and mechanical damage were suggested as clinical symptoms of heavy infestation [38]. Infestation can induce scratching and itching in animals, and this may impair the condition of the host following secondary bacterial infection [29]. The growing problems caused by *Lipoptena* species in Scandinavia, especially in Finland, have been documented [23,27]. In some cases, serious disease symptoms and health problems have been noted in humans, including chronic dermatitis [39] and occupational allergic rhinitis and conjunctivitis [40]. In turn, pricks of another Hippoboscidae representative, *Hippobosca equina*, may be a cause of anaphylactic shock in humans [41]. The situation is additionally complicated by the lack of repellents operating effectively against *Lipoptena* species.

The direct effects of parasitic arthropod bites (i.e., changes caused by the arthropod itself, e.g., through the action of salivary components or mechanical damage to the skin by the mouth organs) may include local inflammatory reactions or chronic skin inflammation [29]. Skin lesions evolving after *Lipoptena* bites are painful and often lead to the development of inflammation of the skin, characterized by the presence of characteristic clots accompanied by pruritus [39] (own observations). What is important in the aspect of health protection, due to the short time of the parasitism of *Lipoptena* spp. on humans and an initially small trace at the bite, is that symptoms reported by the patient are often not associated with infection transmitted by these species [29]. In humans and animals infected with *Bartonella schoenbuchensis*, moderate to severe skin inflammation can develop with a change that can persist for up to a year [29,39,42,43].

Employees of national parks, forest districts, border guard units, members of hunting associations, researchers, field employees of national environmental service institutions, naturalists, or tourist guides are particularly exposed to infestation of ectoparasites and dermatitis [44,45]. Permanent and numerous Hippoboscidae attacks can therefore hinder forestry work and reduce the recreational value of the area and limit its side-use (harvesting forest fruits, mushroom picking, and recreational activities e.g., sport) [4,44]. Companion animals such as dogs in forest areas are also exposed to *Lipoptena* attacks, in particular hunting dogs. Parasites may stay on their skin for a long time, causing inflammatory changes [42,45].

## 5. Conclusions

Considering the health consequences of deer ked bites related to the transmission of zoonotic agents, the knowledge of this subject should be disseminated among people who are exposed to their attacks. Both veterinarians and healthcare professionals should be aware that their animal and human patients should be monitored for signs of disease after deer ked bites.

## Figures and Tables

**Figure 1 insects-11-00859-f001:**
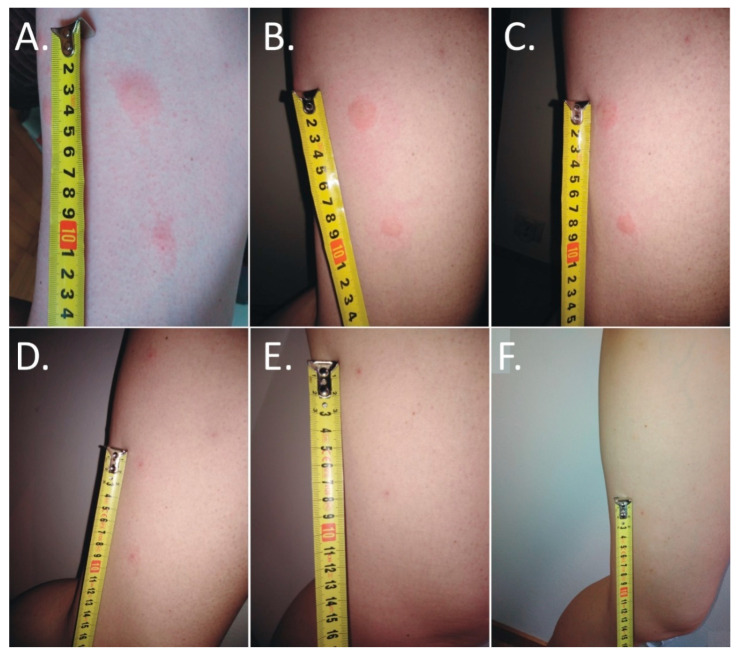
Evolution of skin lesions after deer ked (*Lipoptena cervi*) bites. (**A**,**B**) Multiple painful urticarial pink papules on an erythematous base on the thigh: (**A**) irregular one day after the bite; (**B**) more regular two days after the bite); (**C**) excoriated erythematous papules with self-limited dermatitis (three days after the bite); (**D**) hyperpigmented papules on an erythematous base (5 days after the bite); (**E**) hyperpigmented papules on an erythematous base (4 weeks after the bite); (**F**) residual hyperpigmented papules 6 weeks after the bite).

**Figure 2 insects-11-00859-f002:**
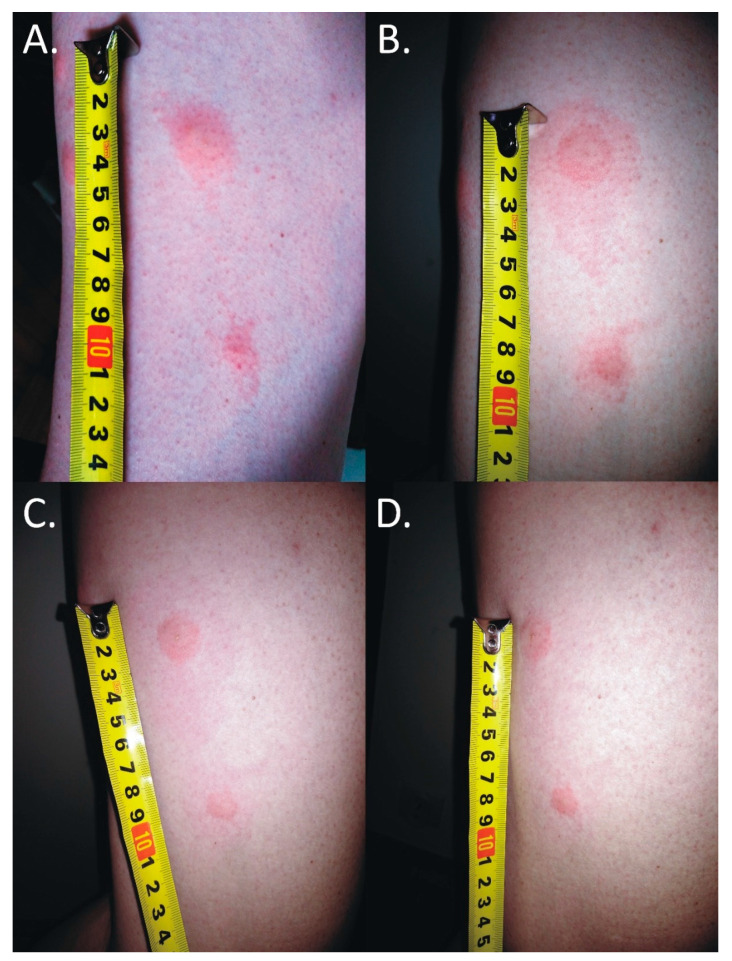
Inflammatory reaction to the deer ked (*Lipoptena cervi*) bite. (**A**) 25 mm × 15 mm erythema 17 h after the bite; (**B**) 50 mm × 25 mm erythema 25 h after the bite; (**C**) 68 mm × 30 mm erythema 40 h after the bite; (**D**) self-limited 50 mm × 30 mm inflammatory reaction 64 h after the bite; the black lines mark the extent of inflammatory changes.

**Figure 3 insects-11-00859-f003:**
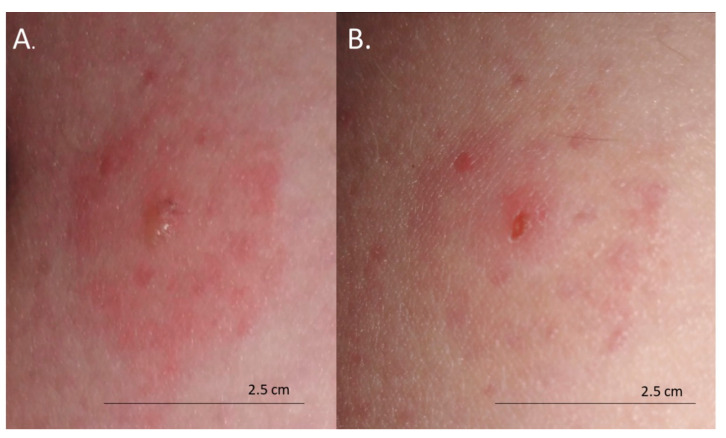
Bites from deer ked (*Lipoptena cervi*): (**A**) formation of vesicles on the skin 48 h after the bite; (**B**) small erosions 72 h after the bite.

**Figure 4 insects-11-00859-f004:**
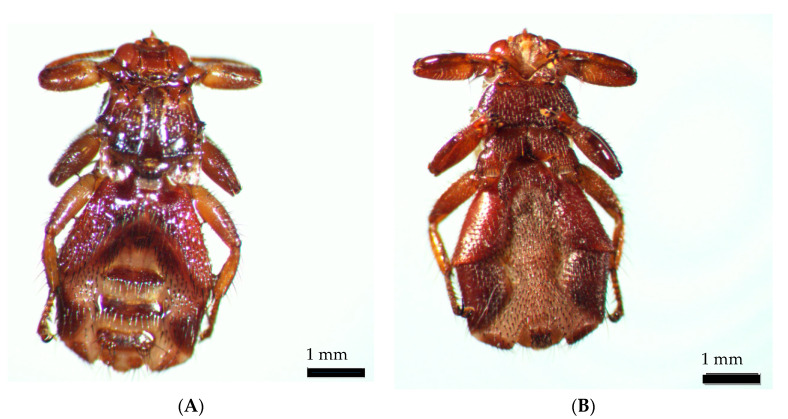
*Lipoptena cervi* female: (**A**) dorsal view; (**B**) ventral view.

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
