# Peer review of "Exposure of Humans to Attacks by Deer Keds and Consequences of Their Bites—A Case Report with Environmental Background"

_insects, 2020, doi:10.3390/insects11120859_

Round 1

Reviewer 1 Report

Weronika Maślanko, Katarzyna Bartosik, Magdalena Raszewska-Famielec, Ewelina Szwaj, and  Marek Asman.  Exposure of humans to attacks by deer keds and consequences of their bites.

The manuscript describes the reaction of a human subject to the bites of a deer ked, Lipoptena cervi. As a case study / observational study, the manuscript is relatively well written. The introduction provides a good background on the biology of deer keds. The case description is thorough and well done. As deer populations continue to increase, human interactions with their parasites will increase as well. The paper provides practitioners and researches with the symptoms to expect from these bites. I believe the manuscript is acceptable for publication as a short note or case study with a few minor editorial changes.

I have noted editorial suggestions and a few comments directly on the manuscript.

Reviewer 2 Report

General comments--This is an interesting case report. The report needs some clarifications.

Specific comments--1. The title does not reflect the study design, a case report. This reviewer suggests the title is changed to indicate that this is a case report; 2. The materials and methods section needs to be expanded to include how the authors find this person. Was this woman a patient of a hospital? Did someone contact you about the case? Otherwise, this patient seems to materialize without origin. Where this happens is explained. The when is not clear. 3. In figure 2, the sharpie lines around the lessons are distracting. You already have a measuring tape. Can you remove them? 4. In figure 4, two different backgrounds are used. Any particular reason? The white background seems to be a better contrast with the tick. 5. What happened to the patient? Was she treated? What treatment did she receive? Besides having the bite site reaction, did she get generalized symptoms? This information is missing.
